# Evolution of a Pathogenic Microbiome

**DOI:** 10.3390/jcm12227184

**Published:** 2023-11-20

**Authors:** Pui Yin Wong, Carmen Yip, Daniel A. Lemberg, Andrew S. Day, Steven T. Leach

**Affiliations:** 1Discipline of Paediatrics, School of Clinical Medicine, University of NSW, Sydney 2052, Australia; rainbow.wong@student.unsw.edu.au (P.Y.W.); carmen.yip@student.unsw.edu.au (C.Y.); andrew.day@otago.ac.nz (A.S.D.); 2Department of Gastroenterology, Sydney Children’s Hospital Randwick, Sydney 2031, Australia; daniel.lemberg@health.nsw.gov.au; 3Department of Paediatrics, University of Otago, Christchurch 8011, New Zealand

**Keywords:** dysbiosis, microbiome, infant, childhood, inflammatory disease

## Abstract

The process of microbiome development arguably begins before birth. Vertical transmission of bacteria from the mother to the infant is a keystone event in microbiome development. Subsequent to birth, the developing microbiome is vulnerable to influence from a wide range of factors. Additionally, the microbiome can influence the health and development of the host infant. This intricate interaction of the gastrointestinal microbiome and the host has been described as both symbiotic and dysbiotic. Defining these terms, a symbiotic microbiome is where the microbiome and host provide mutual benefit to each other. A pathogenic microbiome, or more precisely a gastrointestinal microbiome associated with disease, is increasing described as dysbiotic. This review seeks to investigate the factors that contribute to evolving a disease-causing or ‘dysbiotic’ microbiome. This review covers the development of the gastrointestinal microbiome in infants, the interaction of the microbiome with the host, and its contribution to host immunity and investigates specific features of the gastrointestinal microbiome that are associated with disease.

## 1. Introduction

There are estimated to be approximately one hundred trillion microbes in or on the body, which outnumber human cells at a ratio of 1.3:l [1]. Most of these microbes are located in the gut to form the gastrointestinal (GI) microbiome [2]. This microbiome consists of bacteria, viruses, archaea, and eukaryotes from over 1500 species, contributing to various physiological functions [3]. This microbiome has integral roles in human health, immune development, and nutrient metabolism [4,5,6]. However, alterations in this microbiome can contribute to inflammation and have also been closely associated with the occurrence of multiple diseases [2].

The origin of the gut microbiome is multi-faceted. The very first exposure being the maternal microbiome, which is vertically transmitted during the perinatal period and determines the first colonisers in the infant [7]. Thus, mother-to-infant transfer is an important checkpoint in early life, defining the acquisition and succession of the early-life microbiome [8,9]. Emerging evidence suggests that this process occurs in various patterns, durations, and body sites. It is affected by a multitude of factors, all with the capacity to alter the infant’s microbiome [10,11,12,13]. The recent recognition that the modulation of the early gut microbiome offers great potential to exert substantial changes in health outcomes has attracted great interest towards this field. This review aims to provide an overview of the evolution of the GI microbiome and explores the concept of a pathogenic microbiome.

## 2. Vertical Transmission of the Maternal Microbiome

### 2.1. Transmission of the Placental Microbiome

The placenta is an organ that plays a vital role in the development of the foetus, responsible for the frequent exchange of nutrients, oxygen, and waste products between the mother and foetus [14]. Traditionally, the placenta has been thought of as a sterile environment free of microorganisms. One of the earliest studies supporting the sterility of the placenta, which utilised 16S rRNA sequencing, provided evidence that there was an indistinguishable difference of bacterial species copy numbers between placental biopsies and study controls [15]. This was subsequently corroborated by Lager et al. [16] who utilised 18S rRNA sequencing and detected no eukaryotic signals in placental samples of women with adverse pregnancy outcomes and healthy controls. Several studies also attributed the putative existence of a prenatal microbiome to the contamination of kits and reagents in the laboratory [15,17,18].

However, over the past decade, emerging studies have contradicted the dogma of a sterile placenta. The vertical transmission of bacteria via the placenta is thought to occur through two main routes—haematogenous and ascendant [19]. The former occurs where the microorganisms cross the placenta and colonise the foetal gut, whereas the latter refers to microorganisms resident in the cervix, vagina, or other body sites ascending to the uterus to colonise the foetal gut [19]. In 2020 and 2021, two studies conducted by Rackaityte et al. [20] and Mishra et al. [21], respectively, provided more convincing evidence to support this hypothesis. Microbial signals were consistently detected across different foetal organs including the placenta [20]. Specifically, *Lactobacillus* and *Staphylococcus* were identified as the most prevalent [20]. Experiments conducted in vitro confirmed that bacterial antigens in utero induced the activation of memory T-cells in foetal mesenteric lymph nodes, as opposed to contamination from the external environment [21]. These findings allude to the existence of viable microbes in utero and the extent to which a low-biomass community can promote foetal development before birth. Nonetheless, the mechanisms of vertical transmission through the placenta are not yet fully understood and the contamination in low-biomass samples argues against this possibility. Further studies with rigorous aseptic techniques and novel technologies to improve the sensitivity in the analysis of low-biomass samples is needed to understand the importance of placental microbiome transfer.

### 2.2. Transmission of the Vaginal Microbiome

The mode of delivery is accepted as having a crucial role in early gut microbiome colonisation. During vaginal delivery, the neonate is first exposed to the vaginal and gut microbiome of the mother [10]. The major microbiome that predominates the infant’s gut is hence the maternal vaginal microbiome, with a minor component being the external environment [10,22]. Studies have shown that the maternal vaginal microbiome constitutes up to 16.3% of the infant’s total gut microbiome at day 1, which is not observed in infants delivered by Caesarean section [8].

Several studies have investigated the transmission of the microbiome during delivery, with the consensus that the microbial community of vaginally delivered neonates exhibit a diverse microbiome that is constituted from maternal vaginal and intestinal bacteria [10,23,24]. This is dominated by *Lactobacillus*, *Prevotella*, *Bifidobacterium*, *Parabacteroides*, and *Escherichia* [10,23,24]. Conversely, infants delivered via Caesarean section are found to have increased relative abundances of *Klebsiella*, *Clostridia*, *Enterobacter*, *Staphylococcus*, and opportunistic pathogens, mainly acquired from the maternal skin and hospital environment [8,25,26].

*Bifidobacterium* and *Lactobacillus* are essential in promoting immunogenic humoral and cell-mediated maturation, with immediate roles in the development and protection of the gut microbiome against pathogenic bacteria [27]. However, in Caesarean-delivered infants there can be delays of up to 4 months to acquire key bacteria [26].

The persistence patterns of bacteria in vaginally delivered infants also vary. These infants are enriched with *Lactobacillus*, *Bifidobacterium bifidum*, *Bifidobacterium longum*, and *Parabacteroides diastonis*, and these bacteria are better colonisers as opposed to *Streptococcus salivarius*, *Staphylococcus massiliensis*, *Staphylococcus hominis*, and *Veillonella parvula*, which tend to only be in the gut transiently [8]. A possible explanation is that vertically transferred bacteria are better adapted to the intestinal environment and therefore have a competitive advantage in persisting in the infant’s microbiome.

### 2.3. Transmission of Breast Milk Microbiome

Breast milk accounts for the second most common source of microorganisms after the birth canal for infants that are born vaginally [28]. Studies have proposed that microbes translocate from the maternal gut to the mammary gland, with some influences from the maternal skin and oral cavity during breastfeeding [29,30,31].

The World Health Organisation (WHO) recommends exclusive breastfeeding for at least the first six months of life, following which an introduction to solid foods should commence [32]. It is estimated that 1–6-month-old infants, with an average breastmilk consumption of 670 mL/day [33], consume up to 700,000 bacteria daily, serving as a substantial coloniser of the infant gut [34]. Breast milk possesses a complex and dynamic composition that fulfils an infant’s nutritional requirements and contains certain bioactive components including human milk oligosaccharides (HMOs), antibodies, immune cells, antimicrobial proteins, cytokines, peptides, and lactoferrin [35]. These largely help strengthen the immune system of the newborn and offer protection against disease [35]. Previous research showed that non-breastfed infants are at a substantially increased risk of GI and respiratory infections, atopy, and chronic conditions, substantiating the antimicrobial and immunoregulatory properties of breastmilk and potentially breastmilk-derived bacteria [5,36,37].

It is clear that breastmilk contributes to a significant proportion of an infant’s gut microbiome in the first year of life [38,39,40,41]. Utilising strain-level analyses and metagenomic sequencing, several studies identified shared bacterial strains between maternal milk or faecal samples, and infant faecal samples [38,39,40,41]. Several bacterial strains were consistently identified, with the most prominent being *Bifidobacterium breve* and *B. longum* [38,39,40,41]. Although the sample sizes of these studies were small and negative controls were not included in all the studies, the consistency of the results support associations between breast milk and the development of the infant gut microbiome.

A larger randomised control trial with 107 mother–infant dyads found that the breast milk bacterial communities were significantly associated with the infant gut microbiome during the first 4 months of life [28]. *Bifidobacterium*, *Lactobacilli*, and *Staphylococci*, which dominated the milk of lactating mothers, were observed at an increased abundance and diversity in the stool samples of breastfed infants compared to non-breastfed infants [28]. The number of shared microbes and bacterial diversity also increased with the proportion of breastmilk intake in a dose-dependent manner [28]. Furthermore, each infant’s gut microbiome resistome was found to be more similar to their respective mother’s microbiome than that of an unrelated woman [42,43].

This evidence suggests that breastmilk supports the growth of organisms in the infant’s gut, and acts as a potential site of microbial transmission between mothers and infants. However, data surrounding the proportion of shared microbes should be interpreted with caution, as it remains unclear whether the bacterial communities identified in the infant stool samples are derived from sources other than breastmilk. A primary limitation of these studies was the taxonomic resolution, which limited identification of the bacteria to the genus or species level; these genus or species are commonly found at various body sites and across different individuals. Furthermore, most of the studies investigating the vertical transmission of breast milk also contained limited numbers of dyad pairs and limited culturing of targeted microbes. As such, further studies with larger sample sizes and strain-level analysis will assist in overcoming previous limitations and provide more insight into the role of breast milk in the development of the infant’s microbiome.

### 2.4. Other Potential Sources of Transmission

The other potential sources of vertical transmission include the maternal oral cavity and skin. The relative abundance of shared species in the oral cavity and skin accounted for approximately 7% and 5% of the infant’s microbiome, respectively [8]. They were reported to be more pronounced for up to the first three days of life but decreased with time, suggesting that they are transient colonisers of the gut [8]. Though these sites are not significant contributors to the infant’s initial bacterial community, they may have an influence in later life, opening an avenue for further research.

## 3. Development of the Gut Microbiome from 0 to 6 Months

Within 24 h of birth there is a high diversity of species observed in the infant faecal microbiome, reflecting the rapid influx of pioneering microbes from multiple sources. Diversity and richness then rapidly decrease as unsuitable or poorly adapted organisms are lost or replaced [8,44]. Common species observed at day 1 but not later include: *Clostridium innocuum*, *Streptococcus parasanguinis*, *Prevotella melaningonica*, *Alistipes putredinis*, and *Haemophilus parainfluenzae* [44]. This pattern suggests an influx of species stochastically, followed by the establishment of a core set of species that persists stably in the gut. Of interest, Chu et al. [23] studied 117 stool samples and identified *Klebsiella* and *Escherichia* as the genera with the highest specificity to the neonate’s gut. Notably, these taxa are known facultative anaerobes that are typically observed in the early GI tract, as the newborn gut is predominantly an aerobic environment at birth.

In the weeks following birth, the GI environment becomes anaerobic, thereby favouring the colonisation of strict anaerobes. This is observed by the prevailing taxa in the infant’s gut changing from mostly facultative anaerobes to strict anaerobes—namely *Bifidobacterium*, *Clostridium*, and *Bacteroides* [38,45,46]. At this stage, when the primary source of an infant’s nutrition is breastmilk, the infant’s gut favours the propagation of certain microbes such as *Bifidobacterium* that selectively metabolise HMO in breastmilk. This provides *Bifidobacterium* a nutrient source unavailable to other colonisers including potential pathogens, allowing it to thrive in the developing microbiome [9,47]. *Bifidobacterium* and HMO act synergistically to provide optimal protection to the infant due to their immunogenic roles in preventing the attachment of pathogens, suppressing inflammation, and promoting immune development [47]. In a healthy, vaginally delivered, full-term and breastfed infant, *Bifidobacterium* is the dominant bacteria 1–2 weeks after birth. However, *Bifidobacterium* is then reduced at 4–6 months of age with weaning and after the addition of solid foods to the infant diet as other environmental factors continue to alter the gut microbiome [8,45].

With the introduction of solid food and cessation of breastfeeding there are increased abundances of *Bacteroides* and *Clostridiales* [8,45]. This is associated with an increase in the functional capacity of the microbiome specifically in its capacity to degrade complex sugars, starch, and increase vitamin biosynthesis [8,45]. The explanation for this change is attributed to infant diet modification, as this new diet consists of many polysaccharides, which their immature digestive system cannot digest, thereby trigging an increase in these microbes to facilitate this process [48]. From this point, the alpha diversity and composition of the infant gut microbiome continues maturing until five years of age, and steadily begins to resemble the adult-like structure [8,49].

It is also important to note that different geographical locations exhibit variations in gut microbial composition. In a study by Kuang et al. [50] Proteobacteria, particularly Enterobacteriaceae, were found to be the dominant type of bacteria in Chinese infants <3 months of age. In contrast, Actinobacteria, especially *Bifidobacterium*, were found to be the dominant genera of the intestinal microbiome of Western infants, with a smaller bacterial diversity compared to infants in developing countries [50]. Moreover, Enterobacteria were present in Pakistani and Chinese infants during the first weeks of life while they were only detected in Swedish infants at 6 months of age [51,52].

The variability in the rate of acquisition and gut microbiome profiles are hypothesised to be driven by many factors including host genetics [53], sanitation levels [52], variations of vaginal and skin microbial composition in mothers [52], and higher formula feeding rates in Western and other upper-middle income countries [54]. It is thus evident that regional variances contribute to the interindividual compositional differences of an infant’s microbiome. Nevertheless, it is accepted that by around the age of five years, the infant microbiome has essentially developed and fulfills the functions of an ‘adult’ microbiome.

## 4. Composition and Function of the Developed Microbiome

In a healthy adult, the gut microbiome is predominated by the phyla Firmicutes, Bacteroidetes, Actinobacteria, Proteobacteria, Fusobacteria, and Verrucomicrobia [55,56]. The adult intestinal microbiome also contains a gene set approximately 150 times larger than that of the human genome [55,57]. Through symbiotic interactions with host structures, this microbiome participates in metabolic and immunological processes such as the digestion of nutrients and drugs, synthesis of vitamins, and immunomodulation to maintain homeostasis [3,58,59,60].

However, the specific composition of the gut microbiome varies widely between healthy individuals [61]. Therefore, to understand the gut microbiome requires more than defining set quantities of specific species of microbes, but requires an understanding of the microbiome functional capacity [61,62]. Overall, a healthy microbiome exhibits diversity, stability, and an ability to fulfil its metabolic activities [63,64]. The richer and more diverse the microbiome, the more resilient it is to external threats, providing an inertia to maintain equilibrium [55,61]. A healthy and functional microbiome also promotes competitive exclusion, preventing the overgrowth of pathogenic bacteria [58].

## 5. The Microbiome and Host Immunity

The microbiome plays an integral role in host immunity. The initial line of defense in host immunity is the intestinal epithelial barrier, which protects the host via a mucous layer against luminal microbes [65]. The microbiome can assist the host in maintaining tolerance to beneficial commensal bacteria by preventing the overgrowth of pathogenic strains [2]. The microbiome can also induce the host to synthesise antimicrobial proteins [3,59] and express local immunoglobulins, specifically secretory immunoglobulin A (IgA) [58].

The microbiome also aids the maturation of the innate and adaptive immune responses. It guides the development of lymphoid tissues and immune cells [58], affecting the movement of neutrophils and the division of T cells, especially regulatory T (Treg) cells to prevent aberrant inflammatory responses [2]. In a healthy gut, innate immune cells such as macrophages and dendritic cells capture luminal antigens and maintain tolerance by activating Treg cells rather than initiating a proinflammatory state [66]. These innate immune cells also migrate to lymphoid tissues to guide the adaptive response [66]. Immune defects have been found in germ-free (GF) mice where there is an absence of bacterial exposure. These defects include impaired development of gut-associated lymphoid tissues and Treg cells and reduced IgA-producing B cells [67,68].

The metabolic actions of the microbiome also contribute to immunity. Bacteria can assist in deriving energy from the fermentation of indigestible carbohydrates into short chain fatty acids (SCFAs), including acetate, propionate, and butyrate [69]. SCFAs can prevent the accumulation of toxic waste products [2,58] and provide chemo-protective properties for the intestinal barrier [3]. They maintain hypoxia, produce antimicrobial compounds, and promote anti-inflammatory cytokines. SCFAs also maintain the intestinal barrier [70] by promoting Treg cell development and enhancing mucous production from goblet cells [71].

However, a damaged barrier can allow bacteria and toxins to translocate into the systemic circulation [65], although the overall impact of translocated bacteria and toxins is dependent of the composition and types of bacteria in the intestinal luminal microbiome.

## 6. Characterising A Pathogenic Microbiome

### 6.1. Dysbiosis

Deviation from a healthy microbiome is commonly described as dysbiosis [66]. Dysbiosis can be defined as an altered composition and diversity of the gut microbiome that is associated with ill-health and disease. More specifically, dysbiosis disrupts immune suppressing host–microbe interactions and promotes inflammation, pathogenicity, and disease. With prolonged dysbiosis, there is an overgrowth of pathogenic bacteria with virulence factors, decreased regulatory or beneficial bacteria, and loss of bacterial diversity [72], which leads to dysfunction in the immune response and the promotion of a proinflammatory environment [65,66].

With dysbiosis, the homeostatic bacterial metabolic functions are altered, which can lead to a defective mucosal barrier [65]. It has been proposed that an increase in mucolytic bacteria such as *Ruminococcus gnavus* [73] and reduction in antimicrobial peptides such as Paneth cell-derived alpha-defensins, compromise the integrity of the mucosal barrier and are contributing factors in conditions such as inflammatory bowel disease (IBD) [74]. These factors also increase the risk of microbial overgrowth and bacterial translocation, which can further activate the host immune response via modulation of proinflammatory signalling pathways [65].

Pathobionts, or bacteria that display pathogenic properties opportunistically, can express virulence factors [75]. One example is the release of endotoxins, which act as ligands that bind cell surface receptors on innate immune cells, initiating cell activation [76]. Activated immune cells then upregulate their expression of pattern recognition receptors, which recognise microbe, pathogen, and danger-associated molecular patterns released by microbes and inflammatory cells [66]. In conjunction with reduced immune suppression from commensal bacteria, pathobionts lead to dysregulation in T cell differentiation that promotes the release of proinflammatory cytokines [77]. The resulting inflammatory environment induces host tissue damage and predisposes to diseases including IBD, irritable bowel syndrome, colorectal cancer, and extraintestinal diseases such as obesity and diabetes [78,79,80]. Table 1 outlines examples of several pathobionts and their molecular mediators suggested to contribute to disease.

A number of indices have been proposed to measure and assess dysbiosis. These include large-scale bacterial marker profiling, relevant taxa-based methods, neighborhood classification, random forest prediction, and combined alpha-beta diversity [89,90,91]. However, the most common method to measure and assess dysbiosis is to measure diversity. Alpha diversity describes the amount of unique taxa (richness) and their distribution (evenness) within a microbial community. Beta diversity is used to assess differences in microbial community composition between individuals, such as between those with a disease state and healthy controls [89]. Overall, multiple indices suggest that a single measure fails to address the broad concepts within dysbiosis. Furthermore, these measures cannot distinguish whether dysbiosis is a cause or consequence of disease [59,92,93].

### 6.2. The Pathogenic Microbiome in Inflammatory Disease

A prominent inflammatory disease is IBD, which is a chronic inflammatory condition of the GI tract with episodes of relapse and remission [65]; it is predominantly classified into Crohn’s disease (CD) and ulcerative colitis (UC). CD can affect the entire GI tract with transmural involvement, while UC is characterised by a more superficial inflammation that is confined to the large intestine [94]. The prevalence of IBD has increased in Western countries, with a rising incidence in newly industrialised countries [71]. Up to 25% of patients present with symptoms during adolescence and young adulthood [95]. Children typically develop a more aggressive disease course [96], with detrimental effects on growth, development, and psychosocial wellbeing [71]. The pathogenesis of IBD is multifactorial, with interactions between host genetics, a dysregulated immune response, environmental exposures, and importantly, changes in the gut microbiome contributing to disease [65].

Different factors can alter the intestinal microbiome [70]. Diet, including feeding from birth, is a notable environmental factor that profoundly impacts microbiome composition. For instance, the microbiome of formula-fed infants is characterised by decreased diversity and bacterial richness, which is associated with inflammation [97]. Bolte et al. [98] also found that the Westernised diet, consisting of lower dietary fibre and higher consumption of animal fat, salt, and sugar [99], induces microbiome characteristics that correspond with intestinal inflammation, while plant-based diets were associated with SCFA-producers and lower abundance of pathogenic bacteria. These observations are also supported by Zheng et al. [100] who found correlations between the abundance of several bacterial species and a proinflammatory diet (defined as a diet containing dietary components known to be associated with inflammation). Two further studies have also found associations between IBD and the consumption of high-sugar foods [101,102].

Antibiotic exposure is an additional environmental factor that can induce a loss of health-promoting bacteria and reduced expression of antibacterial agents and immunoglobulins, resulting in the potential to increase susceptibility to infections [55]. Antibiotic use corresponds with an increase in inflammatory cytokines, alteration of insulin sensitivity, and modulation of the metabolism of SCFAs and bile acids [103]. A cohort study conducted by Hviid et al. [104] reported associations between antibiotic exposure during childhood and the development of IBD. Additionally, maternal antibiotic use during pregnancy and infantile antibiotic exposure increases the risk of very early onset IBD [93,105]. Antibiotic use in infants small for gestational age showed an increase in pathogenic species associated with a ‘dysbiotic’ microbiome [13,106]. In addition to antibiotic exposure in premature and small-for-gestational-age infants, other early life factors including Caesarean section delivery and formula feeding increase the risks of asthma, atopy, obesity, type 2 diabetes mellitus, necrotising enterocolitis, and sepsis [5,6,13,36,37,107,108,109].

Significant efforts have been directed at identifying the pathogenic microbiome characteristics of inflammatory disease and in particular IBD [110,111,112,113,114,115,116]. Most studies have identified a decrease in alpha diversity in individuals with CD compared to UC and healthy controls [111,113,114,115,116]. However, a systematic review by Pittayanon et al. [110] reported that the most common findings of a decrease in the species *Faecalibacterium prausnitzii* and an increase in Enterobacteriaceae, were characteristic of IBD. *F. prausnitzii* is a SCFA-producer with metabolomic studies showing reduced SCFAs in the GI tract of individuals with IBD [112].

Furthermore, SCFA-producers, such as *F. prausnitzii* and *Roseburia hominis*, reduce proinflammatory cytokines including interleukin (IL)-12 and interferon-gamma and increase anti-inflammatory cytokines including IL-10 [59], implying that reduced SCFA-producing bacteria may be a marker of the pathogenic microbiome. Enterobacteriaceae, such as *E. coli* and *Shigella*, are facultative anaerobes and are enriched in inflammation at the expense of obligate anaerobes, which may be due to oxidative stress during inflammation [90]. These bacteria produce lipopolysaccharides (LPS), a pathogen-associated molecular pattern that activates Toll-like receptor signalling and a consequent inflammatory cascade [81].

As an illustration of this, a study of GF mice found that the mice developed colitis when colonised by a highly endotoxic intestinal microbiome containing a high proportion of Enterobacteriaceae (including *E. coli*) and a low proportion of Bacteroidetes [117]. Pathogenic strains of *E. coli*, including adherent-invasive *E. coli* (AIEC), have also been reported to be associated with IBD [118]. AIEC can induce the expression of cell adhesion molecules and they possess virulence factors such as type 1 pili and long polar fimbriae, which can facilitate colonisation in the intestinal mucosa [82]. Although *E. coli* can be considered part of the normal GI microbiome, a pathogenic and disproportionate abundance of *E. coli* may also constitute a pathogenic microbiome.

A caveat is that most studies on the microbiome in IBD are cross-sectional or longitudinal over short periods such as one year. Therefore, it is difficult to determine whether the previously described microbiome characteristics are a cause or consequence of disease. Furthermore, the GI microbiome is not homogenous, with reported differences in the microbiome between faecal and biopsy samples [81]. In addition, IBD is a heterogenous disease and findings for individuals with mild disease may not be valid for individuals with moderate or severe disease [111].

### 6.3. The Pediatric Pathogenic Microbiome

A further consideration of the contribution of the microbiome to disease is age. As previously described, in early childhood the gut microbiome is generally considered to be similar in composition and diversity to that of adults [119]. However, some studies report differences in the microbial composition between younger children and adolescents, suggesting continual development of the microbiome with age [62,97,120]. Furthermore, studies of the microbiome of treatment-naïve children with IBD suggests that the microbiome is characterised by reduced abundances of certain bacterial species rather than an increase in pathogens [63,90,121]. This highlights that in children at least, the pathogenic microbiome may be characterised by a loss of physiological functions.

However, a systematic review of gut microbiome profiles of children with IBD, commonly reported decreased alpha diversity and beta diversity that differed from healthy controls [71]. Overall, the IBD microbiome in children showed increased Enterococcus and decreased *Anaerostipes*, *Blautia*, *Coprococcus*, *Faecalibacterium*, *Roseburia*, *Ruminococcus*, and *Lachnospira* [71]. As these findings are also similar to adult IBD, it was proposed that the conditions for adult-onset IBD may be established during childhood [71].

Associations between the microbiome and disease severity or outcomes have also been investigated. Olbjørn et al. [120] reported that patients with higher abundance of Proteobacteria were more likely to require aggressive treatment and surgery. Proteobacteria abundance also correlated with increased CD complications and an absence of mucosal healing [120]. It is of interest that *E. coli* is classified within the Proteobacteria phylum. However, no firm conclusions can be reached whether specific species within the Proteobacteria phylum, or whether a common characteristic shared by the phylum, is contributing to these findings [120].

Paediatric studies have also investigated associations between dysbiosis, disease activity and treatment response [63,71,91,120,121,122,123]. Overall, dysbiosis positively correlated with disease activity in children [71,123]. Of interest was that hydrogen sulfide (H2S)-producers, such as *Fusobacterium*, *Prevotella*, and *Streptococcus*, were increased with inflammation [71]. H2S can damage the intestinal epithelium and influence the microbiome, mucous, and biofilm interactions [71]. Therefore, H2S and H2S-producers should also be a consideration of the pathogenic microbiome.

Contrary to these findings is that de Meij et al. [121] found no association between microbiome profile and disease activity. Further studies have also reported that with treatment, bacterial profiles and dysbiosis remain [91,120]. However, these studies also highlighted that consideration should also be given to the type of treatment, and/or treatment success. Kolho et al. [122] reported that microbial composition and diversity increased in treatment responders, following six weeks of anti-tumour necrosis factor-α therapy. Specific changes associated with treatment response have also been identified with *Eubacterium* and *Bifidobacterium* associated with a favorable response to medication [122,124]. A further study of the oral microbiome also found associations between bacterial species and treatment response [125]. These findings support the idea that analysis of the microbiome may potentially assist in the assessment and management of IBD.

A further factor that requires consideration is that the microbiome itself may be contributing to dysbiosis. The essence of this hypothesis is that symbiont bacteria can potentially evolve into pathobiont bacteria under specific selective pressures [126], which is feasible due to the rapid reproductive cycle of prokaryotes. The potential for a rapidly evolving microbiome is illustrated by the example that there are approximately the same number of reproductive events occurring in the average human microbiome in 5 days, as in the entire human population over the last 66 million years [127].

In recognition of these factors, the concept of microbiome engineering has emerged recently [128]. These include primary preventive strategies such as education on the consequences of Caesarean section delivery, conservative use of antibiotics during pregnancy, administration of antibiotics after cord clamping to limit foetal exposure and the adoption of the WHO Baby Friendly Hospital Initiative [9]. Current secondary prevention methods include pre- and probiotics supplementation of the mother during pregnancy and the neonate after birth, faecal microbiome transplantation and phage therapy [126]. At present, the literature surrounding neonatal microbiome engineering is relatively unexplored but poses a pivotal opportunity to promote robust microbiome development and prevent long-term pathologies associated with a disrupted microbiome.

## 7. Conclusions

Understanding what constitutes a heathy and diseased microbiome is in its infancy. Dysbiosis is increasingly used to describe a disease-causing or ‘pathogenic’ microbiome. However, dysbiosis remains an imprecise concept due to the substantial variability between healthy and diseased individuals, as well as variability in the diseases associated with dysbiosis. Nevertheless, there is some clarity regarding dysbiosis as a general concept. Dysbiosis can be regarded as a microbiome with diminished or lack of normal physiological functions, or one that favors inflammation over immune suppression. In this sense, dysbiosis can contribute to disease manifestation, rendering it pathogenic. Additionally, influential factors that contribute to a pathogenic microbiome are potentially in play even before birth. The importance of mother-to-infant vertical transfer of bacteria to the health of the infant is well-defined and underscores the significance of early microbiome development. Perturbations such as exposure to antibiotics and dietary choices, from birth and throughout life, have clear and measurable effects in altering the microbiome and contributing to dysbiosis.

Overall, the microbiome and its interaction with the host is a complex and highly variable ecosystem. However, investigating the microbiome and its dysbiosis potential (that is the functional, inflammatory, and anti-inflammatory capacity of the microbiome) opens the possibilities for predicting, diagnosing, managing, and potentially preventing inflammatory disease. This suggests a critical avenue for further research and novel interventions in the manipulation of the microbiome for improved human health.

## Figures and Tables

**Table 1 jcm-12-07184-t001:** Examples of pathobionts and their molecular mediators associated with disease.

Pathobionts	Molecular Mediators	Association with Disease
*Escherichia coli*	Produce lipopolysaccharides (LPS), activating toll-like receptor (TLR) signalling and inflammatory cascades including release of interleukin-8 (IL-8) in colonic and intestinal epithelial cells [81] Adherent invasive E. coli strains express cell adhesion molecules to facilitate colonisation in the intestinal mucosa, with virulence factors including type 1 pili and long polar fimbriae expression [82] Promote cancer through activation of T-helper (Th)-17 cells, direct DNA damage, and induction of cholesterol synthesis [78]	IBD Type 2 Diabetes Fatty liver disease Colorectal cancer [3]
*Clostridioides difficile*	Mucolytic enzymes such as cell surface protein Cwp84 which degrades the colonic mucosa and surface layer protein A (SlpA) which activates TLR4-dependent responses and aids colonisation Clostridial toxins A and B (TcdA and TcdB) stimulate inflammatory chemokine and cytokine production, neutrophil influx, disruption of tight junctions, fluid secretion, and epithelial cell death [83]	Pseudomembranous colitis
Enterotoxigenic *Bacteroides fragilis* (ETBF)	Induces colonic signal transducer and activator of transcription-3 (Stat3) activation; colitis characterised by Th-17 response [84] Induces spermine oxidase (SMO) resulting in increased reactive oxygen species (ROS) and DNA damage [85]	Colorectal cancer
*Helicobacter pylori*	Induces SMO leading to ROS production and DNA damage [85] Cytotoxins VacA and CagA injected into host cell, activating oncogenic signal transduction pathways [78] Changes in microbial flora from atrophic gastritis to intestinal metaplasia and dysplasia, leading to increased carcinogenic potential of bacterial strains and formation of carcinogenic N-nitroso compounds [86]	Peptic ulcer disease, gastritis, gastric cancer
*Enterococcus faecalis*	Inoculation of E. faecium strain ATCC 19434 in IL-10 depleted mice increased expression of inflammatory cytokines tumour necrosis factor, IL-1b, IL-6, IL-17a, and IL-12b [87]	Ulcerative colitis
*Proteus mirabilis*	P. mirabilis strains present in mice showed bacterial adherence, invasion, and increased production of pro-inflammatory cytokines IL-18 and IL-1α and cell necrosis Induce pro-inflammatory pathways including nucleotide-binding oligomerisation domain (NOD)-like receptor signalling, Jak-STAT signalling, and MAPK signalling [88]	Crohn’s disease

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
