# Peer review of "Evolution of a Pathogenic Microbiome"

_jcm, 2023, doi:10.3390/jcm12227184_

Round 1

Reviewer 1 Report

Comments and Suggestions for Authors

The manuscript by et al highlights the development of the gastrointestinal microbiome in newborns, the interaction of the microbiome with the host and its contribution to host immunity and analyzes some characteristics of the gastrointestinal microbiome associated with disease.

The authors could read these manuscripts where the role of the microbiota in various pathologies is evident

AÄŸagündüz D, Cocozza E, Cemali Ö, Bayazıt AD, Nanì MF, Cerqua I, Morgillo F, Saygılı SK, Berni Canani R, Amero P and Capasso R (2023), Understanding the role of the gut microbiome in gastrointestinal cancer: A review . Front. Pharmacol. 14:1130562. doi: 10.3389/fphar.2023.1130562

AÄŸagündüz D, Kocaadam-Bozkurt B, Bozkurt O, Sharma H, Esposito R, ÖzoÄŸul F, Capasso R,

Microbiota alteration and modulation in Alzheimer's disease by gerobiotics: The gut-health axis for a good mind. Biomedicine & Pharmacotherapy, 2022,153, 113430

AÄŸagündüz D, Çelik E, Cemali Ö, Bingöl FG, Özenir Ç, ÖzoÄŸul F, Capasso R. Probiotics, Live Biotherapeutic Products (LBPs), and Gut-Brain Axis Related Psychological Conditions: Implications for Research and Dietetics. Probiotics Antimicrob Proteins. 2023 May 24. doi: 10.1007/s12602-023-10092-4.

AÄŸagündüz D, Icer MA, Yesildemir O, Koçak T, Kocyigit E, Capasso R. The roles of dietary lipids and lipidomics in gut-brain axis in type 2 diabetes mellitus. J Transl Med. 2023 Apr 2;21(1):240. doi: 10.1186/s12967-023-04088-5.

In the Discussion, the Authors should highlight the possible clinical significance

Reviewer 2 Report

Comments and Suggestions for Authors

The review may be improved by the following suggestions:

1. The language needs correction at places throughout the manuscript

2. The following may be discussed and referenced:

a. Bliven KA, Maurelli AT. Evolution of Bacterial Pathogens Within the Human Host. Microbiol Spectr. 2016 Feb;4(1):10.1128/microbiolspec.VMBF-0017-2015. doi: 10.1128/microbiolspec.VMBF-0017-2015. PMID: 26999399; PMCID: PMC4804625.

b. Zeng Q, Wu S, Sukumaran J, Rodrigo A. Models of microbiome evolution incorporating host and microbial selection. Microbiome. 2017 Sep 25;5(1):127. doi: 10.1186/s40168-017-0343-x. PMID: 28946894; PMCID: PMC5613328.

c. Henry LP, Bruijning M, Forsberg SKG, Ayroles JF. The microbiome extends host evolutionary potential. Nat Commun. 2021 Aug 26;12(1):5141. doi: 10.1038/s41467-021-25315-x. PMID: 34446709; PMCID: PMC8390463.

d. Venkatakrishnan A, Holzknecht ZE, Holzknecht R, Bowles DE, Kotzé SH, Modliszewski JL, Parker W. Evolution of bacteria in the human gut in response to changing environments: An invisible player in the game of health. Comput Struct Biotechnol J. 2021 Jan 11;19:752-758. doi: 10.1016/j.csbj.2021.01.007. PMID: 33552447; PMCID: PMC7829112.

Comments on the Quality of English Language

The grammar needs improvement at places throughout the manuscript

Reviewer 3 Report

Comments and Suggestions for Authors

Several considerations for this review:

·      The abstract length is too short. Please enlarge it.

·      I am unsure if the authors mean referring to the microbiome instead of the microbiota. Please revise the manuscript in light of these terminological differences.

·      The coverage of the review includes several aspects, from microbiota transmission to dysbiosis. The topics seem to be covered in a very cursory manner. It is not clear whether the authors want to explain how the microbiota is transmitted to the infant or if they wish to clarify how a healthy microbiota becomes "dysbiotic." It seems like two divided reviews, and both topics are treated superficially.

·      The discussion of dysbiosis (the authors' stated purpose for the review) is concise and covers about 2-3 pages. The mention of molecular mediators is quick, summarizing.

·      The absence of figures is a serious shortcoming of this review.

·      The absence of tables encapsulating the pathobionts and molecular mediators necessary for their enrichment in the gut microbiota is a further pitfall.

·      In conclusion, what does this review add to the abundant body of reviews already published?

Reviewer 4 Report

Comments and Suggestions for Authors

The review on evolution of a pathogenic microbiome is a well-articulated article. The authors have referenced recent research articles and have provided a detailed insight of the field. 

1. The author needs to mention about the mechanism of action of the dysbiosis, that arises due to inflammation, increased pathogenicity of the microbiota and change in host-microbe interactions. It would be good to see a paragraph on the proposed/suggested mechanism of action. As this would enhance the complete knowledge of microbiome while reading the paper.

Reviewer 5 Report

Comments and Suggestions for Authors

The article "Evolution of a Pathogenic Microbiome" offers a comprehensive and insightful exploration of the concept of dysbiosis within the gastrointestinal microbiome. The authors have skillfully integrated research findings from multiple disciplines, presenting a well-rounded overview of the subject matter.

One of the commendable aspects of this review is its thorough coverage of the developmental stages of the gastrointestinal microbiome in infants. This section provides a solid foundation for understanding the subsequent discussion on dysbiosis. The authors effectively illustrate the dynamic nature of the microbiome, emphasizing its vulnerability to perturbations that can lead to dysbiotic states.

 Furthermore, the discussion on the interaction between the microbiome and the host, and its contribution to host immunity, is exceptionally well-articulated. The article highlights the intricate and reciprocal relationship between the microbiome and the host's immune system, shedding light on the critical role played by the microbiome in maintaining host health.

The investigation into specific features of the gastrointestinal microbiome associated with disease is a crucial aspect of this review. The authors offer valuable insights into the characteristics that distinguish a pathogenic microbiome, providing a foundation for further research and clinical applications.

The article is commendably structured, with a logical flow of ideas and a clear delineation of key concepts. The abstract provides a concise yet comprehensive overview of the review's scope and objectives, effectively setting the stage for the subsequent discussion.

 In terms of improvement, it would be beneficial to include more specific examples or case studies that illustrate the concepts discussed. Additionally, a brief section summarizing potential therapeutic interventions or strategies to modulate dysbiotic microbiomes could enhance the practical applicability of the review.

 In conclusion, "Evolution of a Pathogenic Microbiome" is a well-crafted review that contributes significantly to our understanding of dysbiosis in the gastrointestinal microbiome. Its meticulous examination of the developmental, interactive, and disease-associated aspects of the microbiome provides a valuable resource for both researchers and clinicians in the field. This review is a commendable addition to the existing body of knowledge on microbiome-related diseases.

Comments on the Quality of English Language

Nothing to add minor proofing needed.

Round 2

Reviewer 2 Report

Comments and Suggestions for Authors

The authors have made most of the corrections

The following are not experimental models and are similar to the study conducted by the authors. They maybe discussed and referenced:

a. Bliven KA, Maurelli AT. Evolution of Bacterial Pathogens Within the Human Host. Microbiol Spectr. 2016 Feb;4(1):10.1128/microbiolspec.VMBF-0017-2015. doi: 10.1128/microbiolspec.VMBF-0017-2015. PMID: 26999399; PMCID: PMC4804625.

b. Venkatakrishnan A, Holzknecht ZE, Holzknecht R, Bowles DE, Kotzé SH, Modliszewski JL, Parker W. Evolution of bacteria in the human gut in response to changing environments: An invisible player in the game of health. Comput Struct Biotechnol J. 2021 Jan 11;19:752-758. doi: 10.1016/j.csbj.2021.01.007. PMID: 33552447; PMCID: PMC7829112.

Reviewer 3 Report

Comments and Suggestions for Authors

The manuscript has been revised.

Author Response

We thank the reviewer and note that their comment was the manuscript was revised